# Metastatic Breast Cancer: Prolonging Life in Routine Oncology Care

**DOI:** 10.3390/cancers16071255

**Published:** 2024-03-22

**Authors:** Rudolf Weide, Stefan Feiten, Christina Waßmann, Bernhard Rendenbach, Ute Braun, Oswald Burkhard, Peter Ehscheidt, Marcus Schmidt

**Affiliations:** 1Praxis für Haematologie und Onkologie Koblenz, 56068 Koblenz, Germany; 2Institut für Versorgungsforschung in der Onkologie, 56068 Koblenz, Germany; s.feiten@invo-koblenz.de; 3Johannes Gutenberg-Universitaet Mainz, Universitaetsmedizin Mainz, 55131 Mainz, Germany; 4Onko Nephro Trier Internistische Gemeinschaftspraxis, 54290 Trier, Germany; 5Onkologische Schwerpunktpraxis Braun und Hünermund, 67061 Ludwigshafen, Germany; onkologie@lusanum.de; 6Onkologische Schwerpunkt-Praxis Worms, 67547 Worms, Germany; 7MVZ Hämatologie/Onkologie, 56564 Neuwied, Germany; 8Klinik und Poliklinik für Geburtshilfe und Frauengesundheit, Universitaetsmedizin Mainz, 55131 Mainz, Germany; marcus.schmidt@unimedizin-mainz.de

**Keywords:** metastatic breast cancer, overall survival, routine care, life prolongation

## Abstract

**Simple Summary:**

Breast cancer patients whose tumor has already developed metastases are now living longer and longer in studies. These results are encouraging but must be viewed with caution because study participants are in most cases very special, i.e., they are younger or have fewer concomitant diseases than the average patient. For this reason, we analyzed all patients who were diagnosed with metastases between 1995 and 2022. The analysis showed that survival has also improved continuously in this real-world patient group. In a comparison of the periods 2018–2022 vs. 1995–2000, patients live about 1.5 years longer, a total of approximately 48 months.

**Abstract:**

Overall survival (OS) of patients with metastatic breast cancer (MBC) has improved within controlled clinical trials. Whether these advances translate into improved OS in routine care is controversial. We therefore analyzed retrospectively unselected female patients from five oncology group practices and one university outpatient clinic, whose initial diagnosis of MBC was between 1995 and 2022. A total of 1610 patients with a median age of 63 years (23–100) were evaluated. In all, 82.9% had hormone-receptor-positive disease, and 23.8% were HER2-positive. Evaluation in time cohorts by initial MBC diagnosis date showed a continuous prolongation of median OS from 31.6 months (0.5–237.3+) (1995–2000) to 48.4 months (0.4–61.1+) (2018–2022) (*p* = 0.003). Univariable analyses showed a significant dependence on the time cohort of diagnosis, metastatic status at initial diagnosis, age at metastasis, hormone and HER2 status, general condition, metastasis localization, and the number of affected organs. A multivariable analysis revealed a significant dependence of survival probability on receptor status, general condition, and number of metastatic sites, as well as the time between initial breast cancer diagnosis and the diagnosis date of MBC in months. In sum, OS of patients with MBC has improved continuously and significantly in routine care over the last 27 years.

## 1. Introduction

Results from randomized controlled trials (RCTs) show an increase in survival time in metastatic breast cancer (MBC) [1,2,3,4,5]. However, these results should be evaluated with caution, as study patients represent a highly selected collective. Transferability to routine care is difficult, especially when it comes to answering the question of whether an increase in survival time has been achieved in everyday care over the last 20 to 30 years.

The evaluation of registry data does not provide a uniform picture. The Munich Cancer Registry reported in 2005 that there had been no improvement in survival between 1980 and 2000 [6]. Meta-analyses reported contradictory results: one found no life extension between 1980 and 1990, but a significantly longer survival time from 1990 onwards in all biological subgroups, even in triple-negative tumors [7]. The authors explained the longer survival time with the introduction of new drugs, such as paclitaxel in 1992, or other new chemotherapeutic agents, antihormonal therapies and anti-HER2 therapies, all of which were approved after 1992 [7]. A long-term observational study from Sweden confirms this data [8].

The questionable validity and quality of the raw data, the incompleteness of the data set from the initial diagnosis of MBC to death and the often-incomplete representation of therapies and therapy lines are uncertainties and difficulties that make the evaluation and interpretation of registry data difficult. To answer the question of whether a prolongation of life could be achieved in routine care over the last 27 years, we analyzed all patients with MBC in time cohorts.

## 2. Materials and Methods

Retrospective analysis of diagnosis and treatment data of all MBC patients treated between 1995 and 2022 in five oncology practices in Koblenz (1995–2022), Trier (2012–2017), Ludwigshafen (2012–2017), Worms (2012–2017), and Neuwied (2012–2017), as well as in the gyneco-oncology outpatient clinic of the University of Mainz (2018–2022), was carried out. There was explicitly no selection; every breast cancer patient treated in the respective period was included. Data on diagnosis, treatment and disease progression were transferred from the patient files into a database and analyzed using SPSS version 29.

Only anonymized routine data were collected, which is why patient consent was not required. A positive vote from the Rhineland-Palatinate Ethics Committee was available (application number 2022-16441).

Multivariable and univariable survival analyses were performed. Continuous variables were categorized to allow group comparisons. For the following groups, median survival was calculated using the Kaplan–Meier method and differences in survival were tested for statistical significance using log-rank tests: year of diagnosis of metastasis (time cohorts), comorbidities (age adjusted Charlson comorbidity index (aaCCI)), age at diagnosis of metastasis, receptor status, time between breast cancer diagnosis and metastasis in months, and number of metastatic localizations. An adjustment for multiple testing was performed.

The focus of the statistical analyses was on univariable comparisons. The multivariable method was not used for modeling, as the variables investigated are not independent of each other, which is why their effects overlap. For example, the year of initial diagnosis of MBC correlates with the development of new targeted therapies and thus also with the receptor status. For the sake of completeness, however, the results of the multivariable analysis are also reported.

All parameters found to be influential in the univariable analyses were tested in a multifactorial Cox regression. The following parameters were included in forward stepwise modeling: receptor status, metastasis localization, age at initial diagnosis MBC, year of diagnosis MBC, time between initial diagnosis breast cancer and metastasis in months, and number of metastatic localizations.

In order to assess the development of survival over time, the diagnosis years were summarized as follows: January 1995–December 2000, January 2001–December 2009, January 2010–December 2014, January 2015–December 2017 and January 2018–December 2022. In order to ensure comparability with previously published analyses, the period 1995 to 2000 was chosen for the first cohort. The last cohort, 2018–2022, resulted from an individual project that was carried out during this period. The three other cohorts were formed in such a way that they contained a similar number of patients to that of the last cohort.

For these groups, the median overall survival was calculated as the primary study objective and tested for statistical significance. A secondary study objective was overall survival in different biological subtypes.

## 3. Results

### 3.1. Patient and Tumor Characteristics

A total of 1610 patients were retrospectively evaluated; patient and tumor characteristics are shown in Table 1.

The comparison of patient and tumor characteristics showed differences within the five time cohorts. In the comparison 1995–2000 versus 2018–2022, there were more older patients, more patients with a higher aaCCI, and more patients with a reduced general condition ECOG >1 (2.2% versus 17.0%). Furthermore, there were more patients with more than two metastatic sites (6.1% versus 22.6%), more patients with visceral or brain metastases (47.0% versus 58.0%) and more triple-negative tumors (8.9% versus 14.4%) in the later cohort.

Median follow-up time was distributed as follows: 1995–2000: 23 years, 2001–2009: 18 years, 2010–2014: 9 years, 2015–2017: 6 years, 2018–2022: 2 years.

### 3.2. Treatment

A total of 1549 patients (96.2%) received antineoplastic therapy. The median number of lines of therapy was three (1–16).

#### 3.2.1. Hormone-Receptor-Positive Tumors

A total of 1286 patients (79.9%) had a hormone-receptor-positive tumor. In all, 1090 (84.8%) were treated with antihormonal therapy. A median of two (1–6) lines of antihormonal therapy were applied. In all, 75.1% were treated with an aromatase inhibitor (AI), 31.6% with fulvestrant, 22.2% received a CDK4/6 inhibitor, 13.5% tamoxifen, and 9.4% everolimus. In the most recent period of 2018–2022, significantly more patients were treated with a CDK4/6 inhibitor (2015–2017: 25.7%, 2018–2022: 77.0%). In addition, this group of patients received chemotherapy (64.7%) with a median number of two chemotherapy lines (1–12). The most common chemotherapeutic agents were taxanes (39.9%), capecitabine (34.0%), anthracyclines (33.8%), carboplatin (8.8%), and eribulin (5.1%); 0.1% received sacituzumab govitecan (SG) within a RCT.

#### 3.2.2. HER2-Positive Tumors

A total of 327 patients (20.3%) had a HER2-positive tumor. Overall, 81.8% were treated with anti-HER2 therapies, 78.7% received trastuzumab, 35.7% pertuzumab, 20.4% TDM-1, 1.9% trastuzumab–deruxtecan, 15.6% lapatinib, and 1.3% tucatinib. A total of 81.8% received chemotherapy, with a median number of two lines of chemotherapy (1–10). In all, 61.5% of patients received taxanes, 32.8% capecitabine, 25.8% anthracyclines, 10.5% carboplatin, and 3.2% eribulin. The median number of antineoplastic therapy lines was three (1–15). There were differences in the frequency of anti-HER2 therapies depending on the diagnosis period: 1995–2000: 72.2%, 2001–2009: 77.3%, 2010–2014: 84.0%, 2015–2017: 84.5%, 2018–2022: 88.2%.

#### 3.2.3. Triple-Negative Tumors

A total of 163 patients (10.1%) had a triple-negative tumor. In all, 97.2% received chemotherapy. A median of two chemotherapy lines were applied (1–9). Overall, 66.2% received taxanes, 57.0% capecitabine, 35.9% anthracyclines, 33.8% carboplatin, and 18.3% eribulin. Additionally, 14.1% received an immune checkpoint blockade antibody. In the last cohort, eribulin was used more frequently (2010–2014: 20.0%, 2015–2017: 19.2%, 2018–2022: 30.2%); capecitabine, on the other hand, was used less frequently (2010–2014: 71.4%, 2015–2017: 46.2%, 2018–2022: 46.6%). In the last cohort of 2018–2022, 44.2% received an immune checkpoint blockade antibody; SG was used in five patients (11.6%).

#### 3.2.4. Triple-Positive Tumors

A total of 238 patients (14.8%) had a triple-positive tumor. In all, 96.6% received systemic treatment. Three therapy lines (1–15) were applied in median. Overall, 78.7% had anti-HER2 therapies, 76.1% received trastuzumab, 35.2% pertuzumab, 17.8% TDM-1, 1.7% trastuzumab–deruxtecan, 13.0% lapatinib, and 1.3% tucatinib. A total of 77.0% had chemotherapy, 57.8% taxanes, 29.6% capecitabine, and 26.1% anthracyclines. In all, 75.2% had antihormonal therapy, 63.9% were treated with an AI, and 17.4% with fulvestrant. The frequency of anti-HER2 therapies depended on the time of diagnosis: 1995–2000: 70.4%, 2001–2009: 68.8%, 2010–2014: 83.9%, 2015–2017: 81.1%, 2018–2022: 85.0%. Chemotherapies were more often used in the early cohorts: 1995–2000: 100.0%, 2001–2009: 81.3%, 2010–2014: 77.4%, 2015–2017: 67.9%, 2018–2022: 67.5%.

Table 2 provides a detailed therapy overview.

### 3.3. Overall Survival

In the overall evaluation period, 63.2% of patients died (1995–2000: 91.5%, 2001–2009: 86.5%, 2010–2014: 63.5%, 2015–2017: 50.6%, 2018–2022: 29.8%). The median OS was 38.0 months (0.2–237.3+). The median survival by time cohort is depicted in Figure 1 and shows a continuous and statistically significant increase in survival time from 31.6 months (0.5–237.3+) (1995–2000) to 48.4 months (0.4–61.1+) (2018–2022) (*p* = 0.003).

The 1-year survival rates show a small but continuous increase over time: 1995–2000: 82.0%, 2001–2009: 85.2%, 2010–2014: 85.5%, 2015–2017: 86.9%, 2018–2022: 88.1%.

The univariable analyses showed significant dependencies on the year of MBC diagnosis (cohorts), metastatic status at initial diagnosis, age, HER2- and hormone-receptor status, general condition, metastatic localizations, and number of affected organs (*p* < 0.001–*p* = 0.018).

Patients with triple-positive tumors lived the longest (52.3 months; 0.9–189.9+), followed by hormone-receptor-positive/HER2-negative (41.1 months; 0.2–181.0) and HER2-positive/hormone-receptor-negative (36.6 months; 0.9–110.0+). Triple-negative patients had the shortest survival time of 19.9 months (0.2–110.0+) (Figure 2).

OS according to biological subgroups and MBC diagnosis cohorts is depicted in Table 3.

The multivariable analysis showed a significant dependence of survival on the time between initial diagnosis of breast cancer and diagnosis of metastasis, receptor status, stage at initial diagnosis, general condition, and number of organ metastases (Table 4). Year of diagnosis and metastatic localizations were not significant in the final step of the model, as other variables, primarily receptor status, provided a significantly higher predictive contribution.

The cause of death could be determined in 802 of 1038 patients (78.8%). The most common cause of death was breast cancer (91.3%), followed by comorbidities (4.9%).

## 4. Discussion

In our retrospective study, we were able to show that OS in MBC has improved in a statistically significant and clinically relevant manner from 31.6 months to 48.4 months between 1995 and 2022. MBC is still an incurable but treatable disease [9]. The main therapeutic goals are to control the disease and delay progression on the one hand and to maintain quality of life (QoL) on the other [9]. The median survival is 2–4 years, depending on the study [6,7,8,10]. The relative 5-year survival rate according to current data from the American Cancer Society is 30% [11]. Only 3% of patients are considered long-term survivors without recurrence. Despite numerous studies carried out at the end of the 20th century and approved drugs (at that time primarily cytostatics and endocrine monotherapies), the Munich Cancer Registry found no increase in survival time in the period 1980 to 2000 [6]. Advances in the molecular characterization of breast cancer now enable increasingly targeted therapies [12]. Current analyses describe an increase in median OS and attribute this to the introduction of new chemotherapeutic agents, antihormonal therapies, and anti-HER2 therapies [7,8].

The most important variables here are receptor status, general condition, and number of metastatic localizations. This does not mean that the diagnosis or treatment period has no influence, but only that other variables, such as receptor status in particular, have a higher predictive power concerning OS. This effect is presumably mediated by the approval of new targeted drugs that continuously improved OS in small steps. Subgroup studies show that the use of AI and fulvestrant in combination with CDK4/6 inhibitors in hormone-receptor-positive tumors and the use of new anti-HER2 therapies in HER2-positive tumors have improved progression-free and overall survival [1,2,3,4]. The use of immune checkpoint blockade antibodies and antibody-drug conjugates (ADCs) also extended progression-free survival (PFS) in subgroups of triple-negative tumors [13,14]. The crucial question of whether the new drugs, whose development and use entail high costs for the pharmaceutical industry and the healthcare system, can extend survival in routine care has not been answered. This is due to the necessary inclusion and exclusion criteria of the RCT, which mean that elderly patients, comorbid patients with reduced organ function or concomitant diseases, and patients with brain metastases or a specific previous therapy are excluded from participation in the studies. This makes it difficult to transfer the results of the RCT to routine care. The evaluation of tumor registry data is also associated with obstacles inherent to the registry [15]. The present analysis of 1610 non-selected patients with MBC from initial diagnosis to death shows that OS has improved continuously over the last 27 years from 31.6 months (1995–2000) to 48.4 months (2018–2022). Looking at the overall group, triple-positive tumors show the longest survival of 52.3 months, followed by HER2-positive and hormone-receptor-positive tumors. Triple-negative tumors show by far the shortest survival with only 19.9 months. Nevertheless, all biological subgroups show an increased OS compared with the period 2001–2009, although the sometimes very small sample sizes and the different lengths of follow-up periods must be taken into account. In comparison with the 1995–2000 cohort, the results are not entirely uniform; selection effects may have played a role here.

### 4.1. Hormone-Receptor-Positive Tumors

The combination of an AI with a CDK4/6 inhibitor is the current first-line therapy standard for postmenopausal patients in metastatic situations [9,12,16]. This recommendation is based on the significantly longer PFS and OS of a combination therapy between an AI and a CDK4/6 inhibitor compared to AI therapy alone [17] or chemotherapy [18]. CDK4/6 inhibitors were approved by the European regulatory authority from 2016 to 2018. This recommendation is already being implemented in routine care in specialized oncology facilities. In the period 2018–2022, 77.0% of all patients in our study were treated with a CDK4/6 inhibitor as part of antihormonal therapy. The reason for the continuously increasing survival in patients with hormone-receptor-positive tumors is most likely multifactorial due to the consistent use of AI plus CDK4/6 inhibitors in the first line, the use of fulvestrant in the second line, and the use of everolimus in the third line. We are hoping for a further improvement with the approval of oral SERDs (selective estrogen receptor downregulators), which have shown an effect even with a mutated estrogen receptor [19,20]. Another reason for the improved survival of patients with hormone-receptor-positive tumors is probably the new chemotherapeutic agents that were approved after 1990 [7]. A further improvement in PFS was shown by the use of the ADC SG in comparison with the oncologist’s choice of treatment in the TROPICS-02 study [21]. In our group, one patient (0.1%) was treated with SG. Approval has now been granted by the European Medicines Agency (EMA) and has led in the meantime to a valuable new treatment option which is used routinely in appropriate patients in our group practice.

### 4.2. HER2-Positive Tumors

The development of trastuzumab and its use in combination with taxane-based chemotherapy has extended median OS from 20.3 to 25.1 months [1]. In the CLEOPATRA study, the combination of docetaxel with trastuzumab and pertuzumab increased median survival from 40.8 months to 56.5 months [22]. In second-line therapy, a new standard was established in the EMILIA study with trastuzumab–emtansine (T-DM1) [23]. The direct comparison in the second-line treatment of T-DM1 with trastuzumab–deruxtecan (T-DXd) in the DESTINY-Breast03 study showed a clear superiority of T-DXd in terms of PFS and OS [24]. New tyrosine kinase inhibitors (TKIs) have enriched the therapeutic options for HER2-positive MBC. Tucatinib in combination with capecitabine + trastuzumab proved superior to the combination of capecitabine + trastuzumab in the third or more line after trastuzumab, pertuzumab and trastuzumab–emtansine anti HER2-therapy in the HER2-CLIMB study [25]. In our collective, 20.3% of tumors were HER2-positive. A total of 81.8% of all HER2-positive patients were treated with anti-HER2 therapies (1995–2000: 72.2%, 2001–2009: 77.3%, 2010–2014: 84.0%, 2015–2017: 84.5%, 2018–2022: 88.2%). The increasing use of anti-HER2 therapies and in particular the steadily increasing proportion of use of more potent anti-HER2 therapies from 2002 to 2022 is very likely the reason for the lifetime extension in HER2-positive patients in routine care [26]. Recently, the DESTINY-Breast04 study showed a significant OS benefit in HER2 low expressing tumors (score 1+ or 2+) when comparing T-DXd with treatment of physician’s choice after one or two chemotherapy lines [27]. We hope that new targeted anti-HER2 therapies (TKI, ADC, bispecific antibodies) will further extend survival.

### 4.3. Triple-Negative Tumors

Triple-negative breast cancer is associated with the worst prognosis. The median survival is 12 to 18 months, and the relative 5-year survival is 12% [28]. In our study, we observed a continuous increase in median survival from 6.8 months (1995–2000) to 20.3 months (2018–2022). In a monocentric analysis in 2014, we were able to show that survival was prolonged in hormone-receptor-positive and HER2-positive tumors, but not in triple-negative tumors [29]. The observed prolongation of survival time in the present study is probably partly due to the sequential use of all chemotherapeutic agents approved since 1990. A total of 97.2% of all patients were treated with antineoplastic agents, with a median number of two palliative lines. In the last two cohorts, more eribulin and less capecitabine were used. Another reason could be the use of immune checkpoint blockade antibodies. A total of 14.1% of all triple-negative patients were treated with an immune checkpoint blockade antibody, 44.2% in the last cohort of 2018–2022.

In the ASCENT study [14], SG was able to show a PFS extension compared to standard therapy, which led to EMA approval in 2021. SG was already used in five patients (11.6%) in the 2018–2022 cohort.

### 4.4. Triple-Positive Tumors

A total of 238 patients were treated with triple-positive tumors. Overall, 75% received antihormonal therapy and anti-HER2-therapy as part of their antineoplastic treatment.

Detection of the complex crosstalk between ER and HER2 is the biological basis to combine antihormonal therapy with anti-HER2 agents [30]. The addition of pertuzumab or lapatinib to trastuzumab + AI was found to be more effective than single HER2-blockade + antihormonal therapy in the PERTAIN and ALTERNATIVE trials [31,32]. Recently, investigators in the sysucc-002 trial randomized patients with triple-positive MBC into first-line trastuzumab plus chemotherapy or endocrine therapy and showed non-inferiority of the trastuzumab–antihormonal regimen [33]. Based on these encouraging data, international and national treatment guidelines suggest anti-HER2 chemotherapy-free regimens for selected patients not suitable for chemotherapy [34].

### 4.5. Patients with Brain Metastases

Brain metastases are associated with significantly shorter survival. The median survival in the largest German registry study BRAINMET is 7.5 months (HER2-positive 13.2 months, hormone-receptor-positive 6.1 months, triple-negative 4.5 months) [35]. In our collective, we observed a median survival of 20.0 months (0.4–110.0+). Subgroups were distributed as follows: triple-positive 62.0 months (1.1–102.8+), hormone-receptor-negative/HER2-positive 25.1 months (1.2–47.8), hormone-receptor-positive/HER2-negative 15.5 months (0.8–69.7+), triple-negative 10.1 months (0.4–110.0+). This comparatively more favorable survival is probably multifactorial, also due to the accumulation of factors for long-term survival in brain metastasis in this small subgroup [35]. The sequential use of all antihormonal therapeutics, anti-HER2 therapeutics, and chemotherapeutics may also have played an important role.

### 4.6. Strengths and Limitations of Our Analysis

The strength of our study is that 1610 unselected patients with MBC from five oncological specialist practices and one university outpatient clinic were analyzed. Almost complete data sets were available for age, comorbidities, general condition, tumor and metastasis characteristics, treatment sequences, course of treatment, and overall survival. This enables survival analyses depending on the time of diagnosis and known prognostic factors.

Our study is limited by the retrospective approach, which entails potential weaknesses in data quality due to the lack of independent control of the raw data. Although repeat biopsies were frequently performed, there was no systematic re-evaluation of pathology. Therefore, the receptor status reflects the methods used at initial diagnosis, and the phenotype of the tumors was not independently confirmed. The time between initial diagnosis and metastasis is likely influenced by the state of the art in treatment at the time and therefore cannot be fully explained by the inherent characteristics of the tumor. Of the 1610 patients, 1002 were treated in one institution between 1995 and 2022. It should be critically noted that data from 1995–2000 are from only one single center, and data from 2018–2022 from two centers. The survival data of all patients were updated at the end of the overall project in April 2022. Follow-up became shorter as time elapsed, which makes a proper interpretation of OS data challenging. For the univariable analyses, cohorts had to be formed which, even if they appear well justified, ultimately remain arbitrary.

## 5. Conclusions

OS of patients with MBC has increased continuously and significantly over the last 27 years in routine care in oncological institutions in Rhineland-Palatinate, Germany. In our opinion, the reasons for this are multifactorial due to more effective chemotherapeutic agents, antihormonal therapy options, and anti-HER2 therapies.

## Figures and Tables

**Figure 1 cancers-16-01255-f001:**
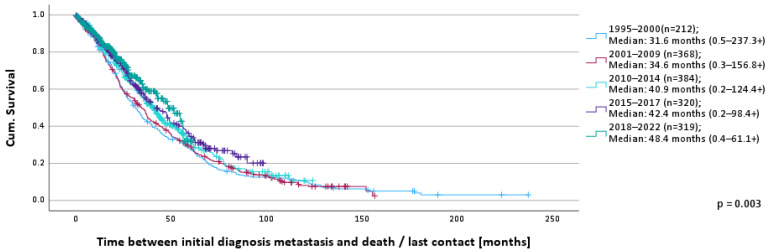
OS according to time of MBC diagnosis.

**Figure 2 cancers-16-01255-f002:**
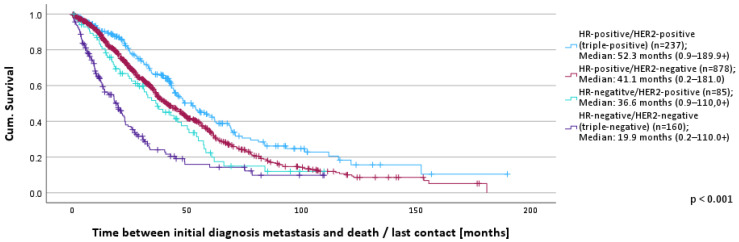
OS according to receptor status.

**Table 1 cancers-16-01255-t001:** Patient and tumor characteristics.

		Total		Year of Initial Diagnosis Metastasis									
				1995–2000		2001–2009		2010–2014		2015–2017		2018–2022	
Age at initial diagnosis metastasis	50 years or younger	315	19.6%	66	31.0%	65	17.6%	63	16.3%	59	18.3%	62	19.4%
	51–60 years	394	24.5%	55	25.8%	105	28.4%	93	24.1%	76	23.6%	65	20.4%
	61–70 years	423	26.3%	52	24.4%	112	30.3%	95	24.6%	78	24.2%	86	27.0%
	71 years or older	478	29.7%	40	18.8%	88	23.8%	135	35.0%	109	33.9%	106	33.2%
ECOG performance status at first presentation	ECOG 0–1	1005	86.1%	90	97.8%	185	89.8%	259	86.0%	207	82.8%	264	83.0%
	ECOG 2–4	162	13.9%	2	2.2%	21	10.2%	42	14.0%	43	17.2%	54	17.0%
Age-adjusted Charlson Comorbidity Index (aaCCI) at first presentation	aaCCI = 6	253	15.8%	57	26.8%	59	16.0%	39	10.2%	46	14.4%	52	16.4%
	aaCCI = 7	346	21.6%	53	24.9%	86	23.3%	77	20.1%	70	21.9%	60	18.9%
	aaCCI = 8	336	21.0%	42	19.7%	94	25.5%	73	19.1%	56	17.5%	71	22.3%
	aaCCI > 8	668	41.7%	61	28.6%	130	35.2%	194	50.7%	148	46.3%	135	42.5%
Grading at first presentation	G1	50	3.1%	7	3.3%	11	3.0%	9	2.3%	10	3.1%	13	4.1%
	G2	768	47.7%	99	46.5%	144	38.9%	197	51.0%	158	49.1%	170	53.3%
	G3	586	36.4%	77	36.2%	166	44.9%	117	30.3%	111	34.5%	115	36.1%
	GX	45	2.8%	19	8.9%	13	3.5%	9	2.3%	4	1.2%	0	0.0%
	not evaluable	161	10.0%	11	5.2%	36	9.7%	54	14.0%	39	12.1%	21	6.6%
Hormone-receptor status (HR)	HR-positive	1286	82.9%	164	86.3%	286	80.8%	311	84.3%	264	82.8%	261	81.8%
	HR-negative	265	17.1%	26	13.7%	68	19.2%	58	15.7%	55	17.2%	58	18.2%
HER2 status	HER2-positive	327	23.8%	36	45.0%	77	26.0%	85	23.4%	77	24.5%	52	16.3%
	HER2-negative	1046	76.2%	44	55.0%	219	74.0%	279	76.6%	237	75.5%	267	83.7%
Receptor status	triple-positive	238	17.4%	27	34.2%	50	17.0%	66	18.3%	55	17.5%	40	12.5%
	HR-positive/ HER2-negative	880	64.4%	36	45.6%	180	61.2%	238	66.1%	205	65.3%	221	69.3%
	HR-negative/ HER2-positive	85	6.2%	9	11.4%	26	8.8%	16	4.4%	22	7.0%	12	3.8%
	triple-negative	163	11.9%	7	8.9%	38	12.9%	40	11.1%	32	10.2%	46	14.4%
Stage at diagnosis of breast cancer	M0	1111	72.9%	174	81.7%	290	78.4%	231	69.4%	193	66.6%	223	69.9%
	M1	414	27.1%	39	18.3%	80	21.6%	102	30.6%	97	33.4%	96	30.1%
Metastatic localizations at initial diagnosis metastasis	Lymph nodes	102	6.3%	19	8.9%	35	9.5%	23	6.0%	14	4.3%	11	3.4%
	bone	544	33.8%	85	39.9%	122	33.0%	116	30.1%	104	32.3%	117	36.7%
	visceral	796	49.4%	96	45.1%	181	48.9%	201	52.1%	157	48.8%	161	50.5%
	CNS	100	6.2%	4	1.9%	17	4.6%	28	7.3%	27	8.4%	24	7.5%
	others	68	4.2%	9	4.2%	15	4.1%	18	4.7%	20	6.2%	6	1.9%
Number of metastatic localisations at initial diagnosis metastasis	1 localization	984	61.1%	159	74.6%	240	64.9%	229	59.3%	198	61.5%	158	49.5%
	2 localizations	381	23.7%	41	19.2%	91	24.6%	95	24.6%	65	20.2%	89	27.9%
	3 and more localizations	245	15.2%	13	6.1%	39	10.5%	62	16.1%	59	18.3%	72	22.6%

**Table 2 cancers-16-01255-t002:** Overview of therapies.

		Total		1st Line		2nd Line		3rd Line		4th Line	
		n	%	n	%	n	%	n	%	n	%
HR-positive/HER2-positive (triple-positive (Total n = 614; 1st line n = 230; 2nd line n = 171; 3rd line n = 125; 4th line n = 88)	docetaxel/trastuzumab/pertuzumab	43	7.0%	34	14.8%	7	4.1%	2	1.6%	0	0.0%
	letrozole	38	6.2%	22	9.6%	9	5.3%	2	1.6%	5	5.7%
	anastrozole	33	5.4%	22	9.6%	6	3.5%	5	4.0%	0	0.0%
	trastuzumab	33	5.4%	1	0.4%	12	7.0%	12	9.6%	8	9.1%
	trastuzumab emtansine	28	4.6%	2	0.9%	12	7.0%	9	7.2%	5	5.7%
	exemestane	25	4.1%	4	1.7%	6	3.5%	11	8.8%	4	4.5%
	letrozole/trastuzumab	24	3.9%	13	5.7%	8	4.7%	2	1.6%	1	1.1%
	(nab-)paclitaxel/trastuzumab	24	3.9%	9	3.9%	5	2.9%	8	6.4%	2	2.3%
	(nab-)paclitaxel/trastuzumab/pertuzumab	20	3.3%	14	6.1%	4	2.3%	1	0.8%	1	1.1%
	fulvestrant	20	3.3%	5	2.2%	4	2.3%	7	5.6%	4	4.5%
	capecitabine	20	3.3%	3	1.3%	6	3.5%	4	3.2%	7	8.0%
	trastuzumab/pertuzumab	20	3.3%	2	0.9%	13	7.6%	5	4.0%	0	0.0%
	letrozole/trastuzumab/pertuzumab	18	2.9%	4	1.7%	12	7.0%	2	1.6%	0	0.0%
	mitoxantrone	17	2.8%	2	0.9%	4	2.3%	5	4.0%	6	6.8%
	epirubicin/docetaxel	16	2.6%	3	1.3%	6	3.5%	6	4.8%	1	1.1%
	vinorelbine/trastuzumab	14	2.3%	3	1.3%	1	0.6%	7	5.6%	3	3.4%
	exemestane/trastuzumab	11	1.8%	6	2.6%	4	2.3%	0	0.0%	1	1.1%
	tamoxifen	11	1.8%	4	1.7%	5	2.9%	1	0.8%	1	1.1%
	epirubicin/cyclophosphamide	10	1.6%	9	3.9%	1	0.6%	0	0.0%	0	0.0%
	docetaxel/trastuzumab	10	1.6%	3	1.3%	3	1.8%	2	1.6%	2	2.3%
	capecitabine/trastuzumab	9	1.5%	2	0.9%	4	2.3%	1	0.8%	2	2.3%
	others	170	27.7%	63	27.4%	39	22.8%	33	26.4%	35	39.8%
HR-positive/HER2-negative (Total n = 2113; 1st line n = 864; 2nd line n = 559; 3rd line n = 404; 4th line n = 286)	letrozole	232	11.0%	156	18.1%	52	9.3%	17	4.2%	7	2.4%
	fulvestrant	230	10.9%	61	7.1%	94	16.8%	42	10.4%	33	11.5%
	exemestane	177	8.4%	62	7.2%	60	10.7%	34	8.4%	21	7.3%
	capecitabine	163	7.7%	25	2.9%	42	7.5%	51	12.6%	45	15.7%
	anastrozole	150	7.1%	114	13.2%	24	4.3%	9	2.2%	3	1.0%
	palbociclib/letrozole	145	6.9%	118	13.7%	18	3.2%	5	1.2%	4	1.4%
	tamoxifen	81	3.8%	46	5.3%	13	2.3%	19	4.7%	3	1.0%
	exemestane/everolimus	81	3.8%	13	1.5%	36	6.4%	26	6.4%	6	2.1%
	capecitabine/bevacizumab	80	3.8%	15	1.7%	27	4.8%	24	5.9%	14	4.9%
	(nab-)paclitaxel	71	3.4%	38	4.4%	13	2.3%	14	3.5%	6	2.1%
	epirubicin/docetaxel	69	3.3%	16	1.9%	18	3.2%	19	4.7%	16	5.6%
	mitoxantrone	68	3.2%	11	1.3%	24	4.3%	20	5.0%	13	4.5%
	palbociclib/fulvestrant	57	2.7%	27	3.1%	18	3.2%	7	1.7%	5	1.7%
	ribociclib/letrozole	33	1.6%	32	3.7%	1	0.2%	0	0.0%	0	0.0%
	vinorelbine	32	1.5%	2	0.2%	7	1.3%	7	1.7%	16	5.6%
	others	444	21.0%	128	14.8%	112	20.0%	110	27.2%	94	32.9%
HR-negative/HER2-positive (Total n = 224; 1st line n = 80; 2nd line n = 65; 3rd line n = 45; 4th line n = 34)	trastuzumab	32	14.3%	7	8.8%	17	26.2%	8	17.8%	0	0.0%
	trastuzumab emtansine	19	8.5%	1	1.3%	8	12.3%	7	15.6%	3	8.8%
	(nab-)paclitaxel/trastuzumab	14	6.3%	9	11.3%	4	6.2%	0	0.0%	1	2.9%
	docetaxel/trastuzumab/pertuzumab	13	5.8%	12	15.0%	0	0.0%	0	0.0%	1	2.9%
	vinorelbine/trastuzumab	12	5.4%	6	7.5%	2	3.1%	3	6.7%	1	2.9%
	trastuzumab/pertuzumab	12	5.4%	1	1.3%	9	13.8%	2	4.4%	0	0.0%
	capecitabine/lapatinib	9	4.0%	3	3.8%	2	3.1%	4	8.9%	0	0.0%
	(nab-)paclitaxel/trastuzumab/pertuzumab	8	3.6%	6	7.5%	2	3.1%	0	0.0%	0	0.0%
	capecitabine	8	3.6%	3	3.8%	1	1.5%	2	4.4%	2	5.9%
	capecitabine/trastuzumab	8	3.6%	3	3.8%	2	3.1%	2	4.4%	1	2.9%
	docetaxel/trastuzumab	6	2.7%	0	0.0%	2	3.1%	2	4.4%	2	5.9%
	(nab-)paclitaxel/carboplatin/trastuzumab/ pertuzumab	4	1.8%	4	5.0%	0	0.0%	0	0.0%	0	0.0%
	others	79	35.3%	25	31.3%	16	24.6%	15	33.3%	23	67.6%
HR-negative/HER2-negative (triple-negative) (Total n = 349; 1st line n = 142; 2nd line n = 105; 3rd line n = 37; 4th line n = 65)	capecitabine	49	14.0%	26	18.3%	13	12.4%	8	21.6%	2	3.1%
	capecitabine/bevacizumab	20	5.7%	20	14.1%	0	0.0%	0	0.0%	0	0.0%
	eribulin	20	5.7%	3	2.1%	10	9.5%	6	16.2%	1	1.5%
	epirubicin/docetaxel	17	4.9%	5	3.5%	7	6.7%	3	8.1%	2	3.1%
	gemcitabine/vinorelbine	17	4.9%	0	0.0%	8	7.6%	3	8.1%	6	9.2%
	vinorelbine	16	4.6%	3	2.1%	5	4.8%	4	10.8%	4	6.2%
	(nab-)paclitaxel	15	4.3%	9	6.3%	4	3.8%	2	5.4%	0	0.0%
	atezolizumab/(nab-)paclitaxel	13	3.7%	11	7.7%	1	1.0%	0	0.0%	1	1.5%
	carboplatin/(nab-)paclitaxel	12	3.4%	5	3.5%	4	3.8%	0	0.0%	3	4.6%
	mitoxantrone	10	2.9%	4	2.8%	4	3.8%	1	2.7%	1	1.5%
	carboplatin/(nab-)paclitaxel/bevacizumab	9	2.6%	7	4.9%	2	1.9%	0	0.0%	0	0.0%
	docetaxel	9	2.6%	5	3.5%	1	1.0%	2	5.4%	1	1.5%
	(nab-)paclitaxel/bevacizumab	9	2.6%	4	2.8%	3	2.9%	2	5.4%	0	0.0%
	clinical trial	8	2.3%	4	2.8%	4	3.8%	0	0.0%	0	0.0%
	carboplatin/gemcitabine	8	2.3%	2	1.4%	3	2.9%	2	5.4%	1	1.5%
	doxorubicin	8	2.3%	0	0.0%	2	1.9%	5	13.5%	1	1.5%
	gemcitabine	7	2.0%	0	0.0%	1	1.0%	4	10.8%	2	3.1%
	cyclophosphamide/methotrexate	6	1.7%	1	0.7%	1	1.0%	2	5.4%	2	3.1%
	epirubicin/(nab-)paclitaxel	6	1.7%	1	0.7%	3	2.9%	1	2.7%	1	1.5%
	others	90	25.8%	32	22.5%	29	27.6%	20	54.1%	9	13.8%

**Table 3 cancers-16-01255-t003:** OS according to receptor status and time of MBC diagnosis.

		Year of Initial Diagnosis Metastasis														
		1995–2000			2001–2009			2010–2014			2015–2017			2018–2022		
		n	Median	Range	n	Median	Range	n	Median	Range	n	Median	Range	n	Median	Range
Receptor status	triple-positive	27	44.9	0.9–189.9+	50	42.5	1.1–156.8+	65	54.0	3.0–112.0	55	84.2	3.0–97.2+	40	48.3	1.6–61.1+
	HR-positive/ HER2-negative	36	60.1	3.3–181.0	179	38.0	0.3–156.7	238	38.9	0.2–124.4+	204	38.9	0.4–98.4+	221	51.6	0.8–60.4+
	HR-negative/ HER2 positive	9	19.1	7.7–84.2	26	27.7	2.8–99.3+	16	46.2	8.0–110.0+	22	47.8	0.9–95.3+	12	-	1.2–53.5+
	triple-negative	7	6.8	1.0–21.2	37	18.4	0.5–109-5+	39	20.2	1.7–110.0+	31	24.5	0.2–78.2+	46	20.3	0.4–43.0+

**Table 4 cancers-16-01255-t004:** Final results of the multivariable analysis.

	Sig.	Hazard Ratio	95% CI for Hazard Ratio	
			Lower Limit	Upper Limit
Time between initial diagnosis breast cancer to initial diagnosis metastasis [months]	*p* < 0.001	0.996	0.994	0.997
Receptor status	*p* < 0.001			
Receptor status: triple-positive vs. HR-positive/HER2-negative	*p* = 0.044	1.278	1.006	1.622
Receptor status: triple-positive vs. HR-negative/HER2-positive	*p* = 0.197	1.307	0.870	1.962
Receptor status: triple-positive vs. triple-negative	*p* < 0.001	2.459	1.801	3.357
Stage at initial diagnosis of breast cancer	*p* < 0.001	0.580	0.460	0.730
ECOG performance status at presentation	*p* < 0.001	1.872	1.468	2.387
Number of metastatic localizations at diagnosis metastasis	*p* < 0.001			
Number of metastatic localizations: 1 localization vs. 2 localizations	*p* = 0.021	1.269	1.037	1.552
Number of metastatic localizations: 1 localization vs. 3 or more localizations	*p* < 0.001	1.883	1.497	2.368

## Data Availability

The data sets used and analyzed during the current study are available from the corresponding author on reasonable request.

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
