# Peer review of "Metastatic Breast Cancer: Prolonging Life in Routine Oncology Care"

_cancers, 2024, doi:10.3390/cancers16071255_

Round 1
Reviewer 1 Report
Comments and Suggestions for Authors
This is an interesting idea to research and the authors tried to prove an increasing overall survival for metastatic breast cancer patients over time.
Nevertheless, the study design has several major flows, such as:
1. Different study populations revealed in the tables, i.e. "1,610 patients were retrospectively evaluated" but other statement is "11,549 patients (96.2%) received antineoplastic therapy". There are multiple inconsistencies in the tables, that lower the value of the declared value.
2. Even if new MBC therapies provided consistent efficacy in RCTs, translated in prolonged OS and PFS, their benefits in daily practice have to be also demonstrated. The authors should elaborate this intuitive finding by matching populations (there are different populations in different periods), offering statistics for each MBC subtype over the prespecified years. The current graphs do not reveal the improvements in each and every subtype, but overall.
3. Discussion is weak in evidence. The literature abundance in RTCs and systematic reviews, even real-world data which were not exploited.
4. The retrospective analysis cannot be a strength and a limitation simultaneously.
In conclusion, I consider the manuscript in this actual form, inappropriate for publication, even if the idea of research is interesting.
Comments on the Quality of English Language
There are multiple English mistakes, not suitable for this type of paper.
Author Response
Dear Reviewer:
We appreciate your comments and have prepared a revised version of the manuscript. Please find enclosed our responses to your comments and suggestions.
Yours sincerely,
R. Weide
Reviewer 1
This is an interesting idea to research and the authors tried to prove an increasing overall survival for metastatic breast cancer patients over time.
Nevertheless, the study design has several major flaws, such as:
Comment 1: Different study populations revealed in the tables, i.e. "1,610 patients were retrospectively evaluated" but other statement is "11,549 patients (96.2%) received antineoplastic therapy". There are multiple inconsistencies in the tables, that lower the value of the declared value. Response 1: The reviewer is right. 11,549 was a typo, which we have corrected. The value is now correct: 1,549
Comment 2: Even if new MBC therapies provided consistent efficacy in RCTs, translated in prolonged OS and PFS, their benefits in daily practice have to be also demonstrated. The authors should elaborate this intuitive finding by matching populations (there are different populations in different periods), offering statistics for each MBC subtype over the prespecified years. The current graphs do not reveal the improvements in each and every subtype, but overall.
Response 2: We agree with the reviewer and have performed statistics for each MBC subtype over prespecified years: HR+/HER2+, HR+/HER2-, HER2-/HR+, triple-negative. The results are shown in table 3.
Comment 3: Discussion is weak in evidence. The literature abundance in RTCs and systematic reviews, even real-world data which were not exploited.
Response 3: We thank the reviewer for this understandably critical comment. To address these concerns, the discussion has been expanded to include triple-positive tumors and more literature has been included. Overall, however, we believe that the discussion and literature should be focused for a publication oriented to clinical reality and brought to the point that is of interest to the reader.
Comment 4: The retrospective analysis cannot be a strength and a limitation simultaneously. Response 4: Once again, the reviewer is right. We apologize for this oversight and have removed it from the manuscript.
In conclusion, I consider the manuscript in this actual form, inappropriate for publication, even if the idea of research is interesting.
Comments on the Quality of English Language
There are multiple English mistakes, not suitable for this type of paper.
Response: We regret this assessment by the reviewer and hope that the revised manuscript is significantly improved.
Reviewer 2 Report
Comments and Suggestions for Authors
The subject is of high interest for clinical practice.
Due to the fact that the type of patients were not similar during all time-points some biasis could appear.
I have some suggestions in coments on the original paper.

Author Response
Dear Reviewer:
We appreciate your comments and have prepared a revised version of the manuscript. Please find enclosed our responses to your comments and suggestions.
Yours sincerely,
R. Weide
Reviewer 2
The subject is of high interest for clinical practice.
Comment 1: Due to the fact that the type of patients were not similar during all time-points some biasis could appear.
Response 1: The reviewer raises an important point. To raise awareness, we have mentioned this possible bias in the manuscript
I have some suggestions in comments on the original paper:
- Comment 2: I am not sure that it is correct to compare for example 1995-2000 having 6 years to 2018-2022 having 5 years.
Response 2: This is indeed a critical point. In the section on methods, we explained how the cohorts were formed. Even if they seem well justified, they always remain somewhat arbitrary due to the retrospective nature of the research and the need to divide into different subgroups. Unfortunately, this is not a prospective RCT that could control for these effects.
- Comment 3: Koblenz has data for all timepoints. Data from four clinics are between 2012 and 2017. No data analized from university clinic before 2018. Could be a bias. Basically for 1995 - 2000 data are from one single center and 2018 - 2022 from two centers.
Response 3: The reviewer is right. We addressed this important point in the discussion section: "For the univariate analyses, cohorts had to be formed, which, even if they appear well justified, ultimately remain arbitrary." As we had to combine the data from different projects, we had to accept a possible bias. Nevertheless, we are confident that the core message of our work is valid.
- Comment 4: time intervals were not equal. Not clear why was selected like this.
Response 4: We agree and have addressed this crucial point in the methods section: "In order to assess the development of survival over time, the diagnosis years were summarized as follows: 01/1995-12/2000, 01/2001-12/2009, 01/2010-12/2014, 01/2015-12/2017 and 01/2018-12/2022. In order to ensure comparability with previously published analyses, the period 1995 to 2000 was chosen for the first cohort. The last cohort 2018-2022 results from an individual project that was carried out during this period. The three other cohorts were formed in such a way that they contained a similar number of patients as the last cohort."
- Comment 5: It seems that there are consistent differences. Are you sure that there are no statistically significant differences?
Response 5: We checked this again, but from a clinical point of view there were no consistent differences between the groups.
- Comment 6: There are different follow-up period.
Response 6: We agree and showed one year survival rate for every cohort.
- Comment 7: [Ref 25] I think HER2-CLIMB is in third or more line (after trastuzumab and TDM1.
Response 7: Here, too, the reviewer is right. We have changed this in the text!
Reviewer 3 Report
Comments and Suggestions for Authors
The manuscript evaluates the survival of metastatic breast patients in one German region. It focuses on whether the year of diagnosis influences the overall survival (OS). The investigation was performed on a relatively large cohort of patients (n=1610) and detailed data are presented. The authors state that OS has improved steadily in the last 27 years. It is an important result and reassures clinicians that the efforts made in the last decades made progress in patient care.
My critical comments:
1. In the result section they mention the differences in patient and tumor characteristics. It seems that in the first cohort, the proportion of young patients is much higher. It may be significant and may explain the differences in ECOG status and comorbidity index. It may be mentioned, too.
2. In Table 1, the “grade 4” needs an explanation, since it does not exist in the present classification.
3. In section 3.2. they mention 11549 patients which is not understandable (the total patient cohort is 1610).
4. I think median follow-up data in different cohorts should be given for the proper interpretation of OS data. Surely, it became shorter as time elapsed.
5. In row 166 they stated “Year of diagnosis and metastatic locations were not included in the final model.” The primary endpoint of the investigation is the effect of the year of diagnosis on OS, therefore, It should be explained why the year of diagnosis was excluded. Without this analysis, they can hardly state in rows 187-188 that “The multivariable analysis clearly shows that the year of MBC diagnosis and the metastasis localizations (even CNS) have no statistically significant influence on OS”.
6. In rows 228-231 they discussed the result of the TROPICS-02 trial. In the specific cohort, only one patient had this treatment, consequently, this sentence seems to be irrelevant and it is better to leave it out of the text.
7. In rows 246-250: “The increasing use of trastuzumab and in particular the steadily increasing proportion of use of more potent anti-HER2 therapies from 2002 to 2022 is very likely the reason for the lifetime extension in HER2-positive patients in routine care.” It should be reflected in the result section, especially "the increasing use of trastuzumab" which is not obvious in the last fifteen years or a citation should be given supporting this statement.
8. In row 210 “Nevertheless, all biological subgroups show a prolongation of life over time compared to 1995 – 2000.” I didn’t find data in the manuscript supporting this statement. In my opinion, it should be added to the results.
9. In rows 253-254 “In our study, we observed a continuous increase in median survival from 6.8 months (1995 – 2000) to 20.3 months (2018 – 2022).” Similarly, detailed data should be given in the result section.
10. Row 258 “The observed prolongation of survival time in the present study is probably partly due to the sequential use of all chemotherapeutic agents approved since 1990.” This should be reflected in the result section if recently they used more chemotherapy than previously.
11. Row 259 “ cytoreductively” may need an explanation.
In my opinion, it should be added in the discussion of the limitations of the study, that
- pathology reevaluation was not performed and therefore, the receptor status of the tumors reflects the method used at the time of primary diagnosis, and the phenotype of tumors has not been confirmed,
- the different follow-up periods in different cohorts may skew the results,
- the time between initial diagnosis and metastasis is probably influenced by the state-of-the-art treatment of its time and therefore, not fully explained by the inherent features of the tumor.
Author Response
Dear Reviewer:
We appreciate your comments and have prepared a revised version of the manuscript. Please find enclosed our responses to your comments and suggestions.
Yours sincerely,
R. Weide
Reviewer 3
The manuscript evaluates the survival of metastatic breast patients in one German region. It focuses on whether the year of diagnosis influences the overall survival (OS). The investigation was performed on a relatively large cohort of patients (n=1610) and detailed data are presented. The authors state that OS has improved steadily in the last 27 years. It is an important result and reassures clinicians that the efforts made in the last decades made progress in patient care.
My critical comments:
Comment 1: In the result section they mention the differences in patient and tumor characteristics. It seems that in the first cohort, the proportion of young patients is much higher. It may be significant and may explain the differences in ECOG status and comorbidity index. It may be mentioned, too. Response 1: We fully agree and have added this to the text.
Comment 2: In Table 1, the “grade 4” needs an explanation, since it does not exist in the present classification.
Response 2: We apologize for this error and have corrected it.
Comment 3: In section 3.2. they mention 11549 patients which is not understandable (the total patient cohort is 1610).
Response 3: The reviewer is right. 11,549 was a typo, which we have corrected. The value is now correct: 1,549
Comment 4: I think median follow-up data in different cohorts should be given for the proper interpretation of OS data. Surely, it became shorter as time elapsed.
Response 4: The reviewer points out an important point. The follow-up data is now displayed and mentioned in the discussion
Comment 5: In row 166 they stated “Year of diagnosis and metastatic locations were not included in the final model.” The primary endpoint of the investigation is the effect of the year of diagnosis on OS, therefore, It should be explained why the year of diagnosis was excluded. Without this analysis, they can hardly state in rows 187-188 that “The multivariable analysis clearly shows that the year of MBC diagnosis and the metastasis localizations (even CNS) have no statistically significant influence on OS”.
Response 5: The reviewer is correct. Accordingly, we have deleted this sentence from the text.
Comment 6: In rows 228-231 they discussed the result of the TROPICS-02 trial. In the specific cohort, only one patient had this treatment, consequently, this sentence seems to be irrelevant and it is better to leave it out of the text.
Response 6: We agree in principle, but for the sake of completeness we have decided not to omit this sentence, but to amend it.
Comment 7: In rows 246-250: “The increasing use of trastuzumab and in particular the steadily increasing proportion of use of more potent anti-HER2 therapies from 2002 to 2022 is very likely the reason for the lifetime extension in HER2-positive patients in routine care.” It should be reflected in the result section, especially "the increasing use of trastuzumab" which is not obvious in the last fifteen years or a citation should be given supporting this statement.
Response 7: We thank the reviewer for this important reference. We have changed the sentence and added a supporting reference.
Comment 8: In row 210 “Nevertheless, all biological subgroups show a prolongation of life over time compared to 1995 – 2000.” I didn’t find data in the manuscript supporting this statement. In my opinion, it should be added to the results.
Response 8: We are in agreement. The data (table) has been added. In addition, the sentence in the discussion has been updated.
Comment 9: In rows 253-254 “In our study, we observed a continuous increase in median survival from 6.8 months (1995 – 2000) to 20.3 months (2018 – 2022).” Similarly, detailed data should be given in the result section.
Response 9: Again, we agree and added detailed data (table).
Comment 10: Row 258 “The observed prolongation of survival time in the present study is probably partly due to the sequential use of all chemotherapeutic agents approved since 1990.” This should be reflected in the result section if recently they used more chemotherapy than previously.
Response 10: In agreement with the reviewer, additional data (results section) have been added.
Comment 11: Row 259 “cytoreductively” may need an explanation.
Response 11: We have changed it into antineoplastic agents.
In my opinion, it should be added in the discussion of the limitations of the study, that
Comment 12: pathology reevaluation was not performed and therefore, the receptor status of the tumors reflects the method used at the time of primary diagnosis, and the phenotype of tumors has not been confirmed
Response 12: This important limitation has been added to the Discussion.
Comment 13: the different follow-up periods in different cohorts may skew the results.
Response 13: We agree with the reviewer and have included this restriction in the discussion.
Comment 14: the time between initial diagnosis and metastasis is probably influenced by the state-of-the-art treatment of its time and therefore, not fully explained by the inherent features of the tumor.
Response 14: The reviewer raises an important point. Indeed, the differences in survival between the different cohorts could also be explained by differences in the optimal treatment at that time point. However, other variables, such as receptor status in particular, have a higher predictive power for survival. This has been added to the discussion section.
Round 2
Reviewer 1 Report
Comments and Suggestions for Authors
The quality of the manuscript improved in this actual version, by detailing OS data for each and every breast cancer subtype.
Language and text mistakes have been corrected.
This article is ready to be published.
Reviewer 3 Report
Comments and Suggestions for Authors
No more comment.